# Study on the Geological Condition Analysis and Grade Division of High Altitude and Cold Stope Slope

Ruichong Zhang [1], Shiwei Wu [1], Chenyu Xie [2,*] and Qingfa Chen [1,*]

1   School of Resources, Environment and Materials, Guangxi University, Nanning 530004, China;
    zhangruichong2002@163.com (R.Z.); gxumining@163.com (S.W.)
2   School of Environment and Resources, Xiangtan University, Xiangtan 411105, China
*   Correspondence: xiechengyu42@xtu.edu.cn (C.X.); chqf98121@163.com (Q.C.); Tel.: +86-182-2948-4803 (C.X.);
    +86-159-7816-9802 (Q.C.)

**Abstract:** Analysis of the geological conditions of high-altitude and low-temperature stope slopes and the study of grade division are the basis for the evaluation of slope stability. Based on the engineering background of the eastern slope of the Preparatory iron mine in Hejing County, Xinjiang, we comprehensively analyse and summarize the factors that affect the geological conditions of high-altitude and cold slopes and finally determine nine geological conditions that affect the index parameters. Based on a back-propagation (BP) neural network algorithm, we establish an applicable network model to analyse the geological conditions of slopes in cold areas. The model is applied to the eastern slope to analyse and classify the geological conditions of the high-altitude and low-temperature slopes. The research results show that the skarn rock layer in the eastern slope is in a stable state and not prone to landslides, and its corresponding geological condition is Grade I; meanwhile, the monzonite porphyry rock layer is in a relatively stable state, with a potential for landslides and a corresponding geological condition Grade II. The marble rock layer is in a generally stable state, there is the possibility of landslide accidents, and the corresponding geological condition level is Grade III. The limestone rock layer is in an unstable state and prone to landslide accidents, it has a corresponding geology condition Grade IV. Therefore, the eastern slope can be divided into different geological condition regions: Zone I, Zone II, Zone III, and Zone IV, and the corresponding geological condition levels for these are Grade I, Grade II, Grade III, and Grade IV. These results may provide a basis for the stability evaluation of high altitudes and cold slopes.

**Keywords:** high-altitude slope; BP neural network; freeze-thaw cycle; geological conditions

## 1. Introduction

Slope instability is one of the world's geological disasters [1–4]. Every year, the economic losses of various countries in the world caused by geological disasters due to slope instability reach immeasurable levels [5–7]. Currently, there are no accurate statistics on the loss, but the loss is still huge. Under the action of freezing and thawing cycles, blasting mining, weathering and other factors, slopes in cold areas can easily cause damage to the mechanical properties of rock slopes and lead to their instability in mines in cold areas [8–11]. Therefore, open pit mine slope landslides are a potential hazard in harsh environments with high altitudes and cold areas. For example, the "329" landslide disaster. On 29 March 2013, a landslide occurred on Zeri Mountain in the Jiama (in Tibet Province) mining area of the China National Gold Group, causing more than 2 million cubic metres of slope landslides. Eighty-three field workers were buried, and the landslide was investigated afterwards. The reason was found to be factors such as the freezing and thawing of ice and snow.

With the implementation of Western development and progress of engineering technology, difficult-to-mine mineral resources and hidden dangers left over by exploited mineral resources in the harsh environments of Tibet, Xinjiang and other cold regions have



begun to receive national attention. The stability of the geological conditions of stope slopes directly affect the smooth development of the mine. Therefore, it is very important to effectively analyse the geological conditions of stope slopes.

The analysis and classification of slope geological conditions can provide an important basis to formulate disaster prevention and mitigation measures and have guiding significance for landslide disaster mitigation planning [12]. Based on the engineering background of the eastern slope of the Preparatory iron mine in Hejing County, Xinjiang, this paper comprehensively analyses and summarizes the factors that affect the geological conditions of high-altitude and cold slopes and finally determines nine geological conditions that affect the index parameters [5,7,13–16]. Based on a back-propagation (BP) neural network algorithm, a network mode is established that is suitable for the analysis of the geological conditions of slopes in cold areas. This model is applied to the east slope to analyse and classify the geological conditions of high-altitude and low-temperature slopes.

## 2. BP Neural Network

### 2.1. BP Neural Network Operation Mechanism

In the early 1980s, the theory and research of artificial neural networks (ANNs) made considerable progress under the influence of the sound and continuous development of computers and the continuous breaking of new technical barriers. The theory of artificial neural networks sprouted during this golden period [17] and flourished as a new research paradigm.

Many studies [18,19] have shown that the efficiency of traditional system theory analysis methods is low, and the scientific nature is slightly lacking. Compared with the traditional system theory analysis method, the analysis method of the BP neural network (back-propagation neural network) is more scientific and efficient, the result is more accurate and can convince most researchers in the process of practical application. Compared with the traditional system theory analysis method, it obviously shows stronger competitiveness.

In addition, many studies [20] have found that the adaptive and self-learning capabilities of a BP neural network are outstanding, and the linear function mapping and nonlinear function mapping problems based on a BP neural network are easier to identify for mining systems.

These reasons have made many researchers in the current scientific research field strongly affirm BP neural networks. A BP neural network is a multi-layer feedforward neural network that must reduce the error between the network output and the actual value to a certain range through repeated training and learning so that the network output can reach the required accuracy. A BP neural network method to reduce the error between the network output result and the actual value trains the propagation algorithm according to the forwards multi-layer network structure and reverse feedback. The flow chart of the learning process of a BP neural network is shown in Figure 1.

### 2.2. Data Processing

This article uses a BP neural network-supervised algorithm to classify the data. The specific method is as follows: by continuously selecting certain characteristic parameters from the sample data that have been collected, trained, checked, filtered, and subsequently set according to the classifier in advance we can determine the criteria and summarize and sort the samples that have been further screened out and have been identified.

Before the data classification process begins, a certain amount of training data must be mastered because the continuous and stable operation of the BP neural network strictly requires the input of relevant training data. Only in this way can it be extracted through the feature extraction of the input training data in the subsequent classification process to establish a scientific and rigorous classification model. Then, the existing training data are analysed by comparing the classification model with the verification model. Finally, the classification of the data is completed.

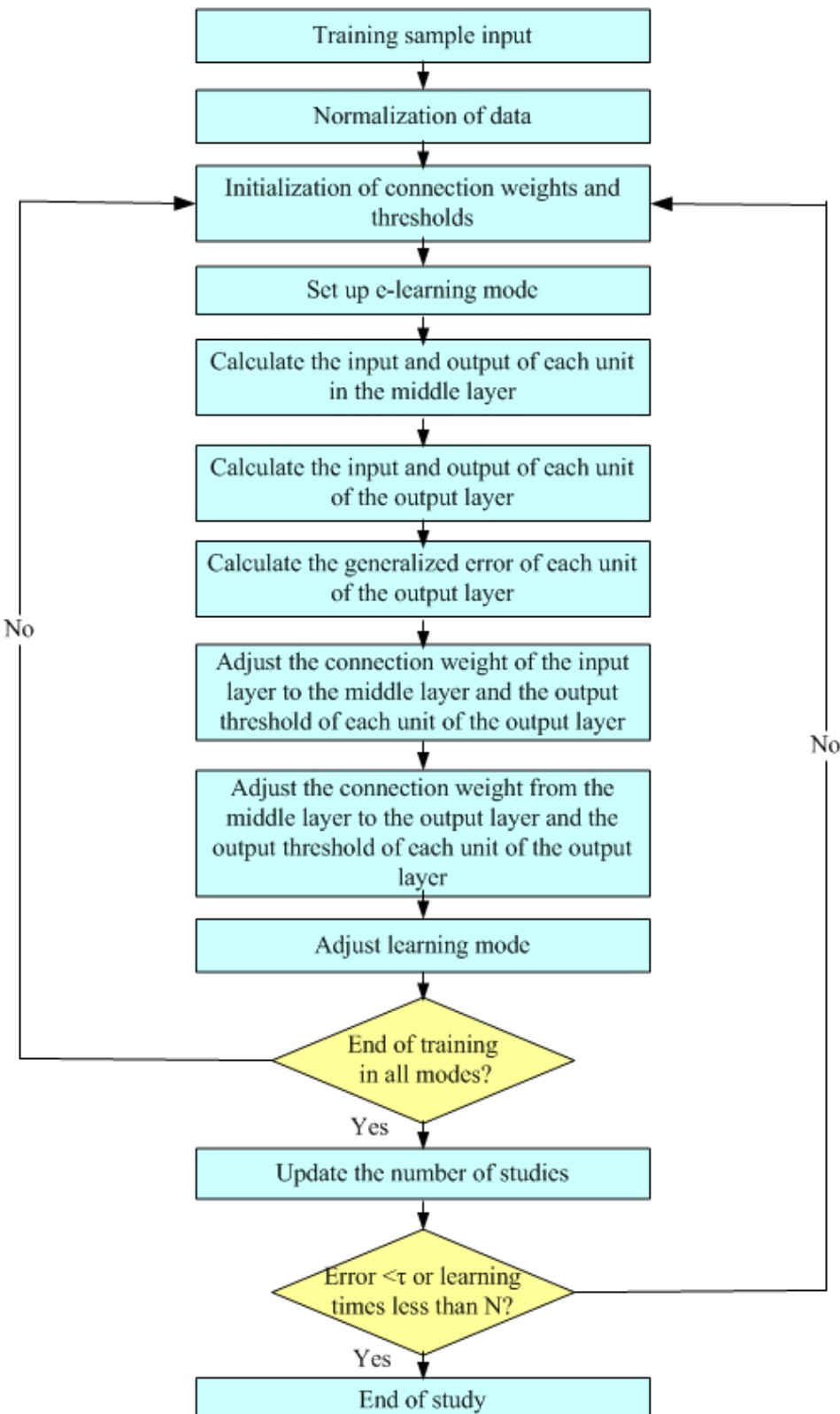

**Figure 1.** BP neural network operation flow chart.

To eliminate the influence of other transformation functions on the transformed image as much as possible, it is necessary to normalize the collected data. Normalization processing refers to the use of the principle of invariant moments to convert the unquantifiable expression form into a value in the range of 0–1 for processing so that the expression form becomes a scalar.

The formula for data normalization is:

$$y = \frac{x - Min_{value}}{Max_{value} - Min_{value}} \tag{1}$$

where $x$ is the value before conversion, $y$ is the converted value, $Min_{value}$ is the sample minimum and $Max_{value}$ is the sample maximum.

The BP neural network actually achieves the accuracy requirements through repeated training of multiple samples to find the minimum value of the error function. The most common method to determine whether the error satisfies the error accuracy requirement is logistic regression. This article uses a binary logistic regression method to determine the two results (True/False) of the input data and the corresponding probability (PTrue/PFalse) of the results to determine whether the accuracy of the network system satisfies the requirements. The formula is as follows:

$$t = wx + b \tag{2}$$

In the formula:
$x$—input sample parameters;
$t$—temporary variable;
$w, b$—model parameters.

The sigmoid function is usually used as the use condition of the conversion function: in the logical judgment, when $h(t) > 0.5$, $y = 1$. The formula is as follows:

$$h(t) = \frac{1}{1 - e^{-t}} \tag{3}$$

From formula (3), we can see: its parameter curve is shown in Figure 2.

### 2.3. BP Neural Network Forward Transmission and Reverse Feedback

(1) Forward transmission

The input parameters of the neural network reach the output end through the input end and each node of the intermediate layer (hidden layer). This method is forward transmission. The intermediate layer can be adjusted by changing the weight relationship between the intermediate layer and the output layer, output threshold updating of the intermediate layer, and other adjustment methods to reduce the generalization error between each node and the actual value, and to therefore achieve the desired result.

The multi-layer perceptron is composed of one or more single-layer perceptrons, which can calculate nonlinear data. The input and output ends of the multi-layer perceptron contain multiple hidden layers [18]. However, thus far, there are different opinions on the number of hidden layers.

The decision-making area of a single-layer perceptron is divided by an extended two-dimensional data plane. In addition, when the multi-layer perceptron contains only one hidden layer, the decision-making area can be an open convex area or a closed concave area. When the multilayer perceptron contains more than one hidden layer, its decision-making area can show diversified shapes and area divisions. Figure 3 shows the change in the weight relationship during the forward transmission.

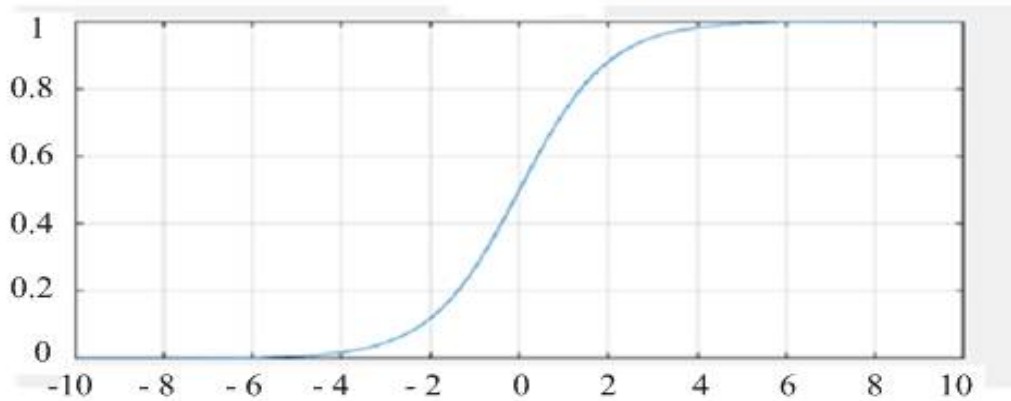

**Figure 2.** The Sigmoid curve.

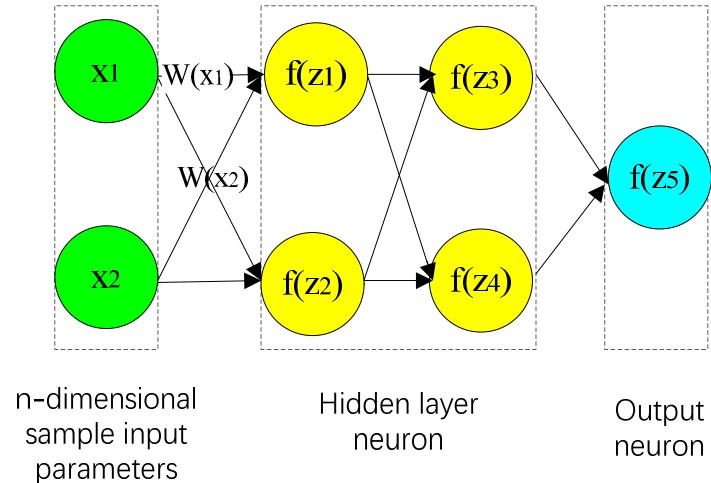

n-dimensional sample input parameters

Hidden layer neuron

Output neuron

**Figure 3.** Graph of the weight relation changes.

Neural network training samples must introduce randomly assigned weights and biases. Simultaneously, the randomly assigned weights and random assigned biases introduced are not randomly selected but must satisfy the weights. The condition of interval real number is (–1, 1), and the offset is (0, 1) interval real number. Only after the abovementioned conditions are satisfied can the network model be forward propagated again. In this process, $X_1$ and $X_2$ are calculated by formulas (4)–(8).

For the neuron $f(z_1)$, the following calculations are performed when only the weight assignment is considered:

$$y_1 = f(z_1) = f(w_{(x_1)1} \times x_1 + w_{(x_2)1} \times x_2) \tag{4}$$

In the formula, $w_{(x_1)1}$ represents the weight of $x_1$ to $y_1$, as shown in Figure 3. Similarly, we can calculate:

$$y_2 = f(z_2) = f(w_{(x_2)2} \times x_1 + w_{(x_2)2} \times x_2) \tag{5}$$

$$y_3 = f(z_3) = f(w_{(y_1)1} \times y_1 + w_{(y_2)1} \times y_2) \tag{6}$$

$$y_4 = f(z_4) = f(w_{(y_1)2} \times y_1 + w_{(y_2)2} \times y_2) \tag{7}$$

$$y_5 = f(z_4) = f(w_{(y_3)1} \times y_3 + w_{(y_4)1} \times y_4) \tag{8}$$

In summary, the output value of each node can be calculated by the formula of forward transmission. Accordingly, the actual output result of the forward transmission model

can also be obtained by calculation, and the final output result $y_5$ obtained by the above formula is exactly the actual output result of the forward transmission mode.

(2) Back feedback

To facilitate the error to participate in subsequent calculations, this article assumes that $t$ is the expected output value of the training data. Because $y_5$ is the actual output value of the forward propagation model, it is assumed that the difference between the actual output value and the expected output value is $\delta = t - y_5$. In addition, the difference between the actual output value and the expected output value must be based on actual conditions during the definition process. It is assumed that there is an error between the actual output value and the expected output value of each node, and this error is defined as $\delta i$. By training the error between the actual value and the expected value, the adjustment of the weight is a crucial step for the feedback adjustment of the BP neural network. The specific process is shown in Figure 4.

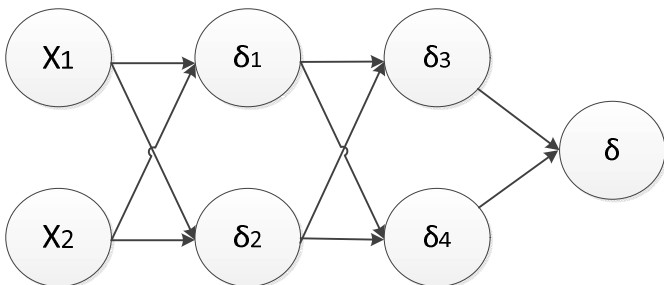

**Figure 4.** Variation diagram of the error feedback relationship.

For the error, $\delta_3 = w_{(y_3)}\delta$, and similarly,. $\delta_4 = w_{(y_4)}\delta$ The calculation method for $\delta_1$ and $\delta_2$ is as follows:

$$\delta_1 = w_{(y_1)1}\delta_3 + w_{(y_1)2}\delta_4 \tag{9}$$

$$\delta_2 = w_{(y_2)1}\delta_3 + w_{(y_2)2}\delta_4 \tag{10}$$

Using formula (9) and calculating according to the principle of this formula, the value of $\delta_1, \delta_2, \delta_3, \delta_4$ can be finally obtained. In addition, the theoretical basis of back propagation is the change in relationship between the error and the weight. The variation $\Delta w_i$ obtained by adjusting the weight is calculated by error. The calculation formula of the weight variation is as follows:

$$\Delta w_i = \eta \delta_i \frac{df(z_i)}{dz_i} x_i \tag{11}$$

where: $\eta$ is the learning rate.

The weight of $w_{(x_1)1}$ can be adjusted as follows:

$$w'_{(x_1)1} = w_{(x_1)1} + \eta \delta_i \frac{df(z_i)}{dz_i} x_1 \tag{12}$$

Similarly, the weight of $w_{(x_2)1}$ is adjusted as:

$$w'_{(x_2)1} = w_{(x_2)1} + \eta \delta_i \frac{df(z_i)}{dz_i} x_1 \tag{13}$$

The weight is calculated and adjusted according to Formula (12), and the final result is an update of the weight. A single back propagation includes the calculation, adjustment and updating of the weights of all nodes. Only after these tasks are completed is back propagation considered completed once. The essence of the realization of the reverse transmission algorithm is to complete the parameter adjustment of the sample model. In this process, forward transmission and reverse feedback are continuously performed. Finally, the error, weight and accuracy of the model reach the desired value.

In summary, the training process of the neural network can be completed through forward transmission and reverse feedback. However, this type of training will not continue indefinitely. Under certain conditions, the training will stop. The BP network training model stops in two situations; after setting and reaching the maximum number of iterations and after reaching a certain threshold.

## 3. Construction of a BP Neural Network Suitable for Preparing Iron Ore Slopes

### 3.1. Geological Condition Analysis and Network Output Parameter Setting

(1) Determine the geological conditions index

In the process of using the BP neural network to classify and predict the geological conditions of the stope slope, it is necessary to establish the corresponding BP neural network model. The first step in establishing a BP neural network model is to evaluate the reliability of its input parameters and filter the parameters to exclude unreliable input parameters so that the final output parameters are as accurate and reliable as possible and can show the influence of different geological factors on the geological conditions of the slope.

Generally, the geological influencing factors of slopes are the slope, slope height, lithology, unit weight, internal friction angle, porosity, cohesion, freeze-thaw cycles, etc. The slope and slope height determine the geometry of the slope and are indispensable factors for its existence. Furthermore, the lithology, gravity, internal friction angle, porosity, cohesion, etc. are important characteristics of the rock mass of the slope as they characterize the quality of the rock mass that composes it. As a unique feature of a slope in a cold region, the freeze-thaw cycle plays a huge role in the classification of geological conditions there. Various geological factors have a certain connection, while some other factors do not. Despite this, all of these factors play a vital role in the division of the slope geological conditions and therefore all will be divided. This is an important factor in the grade of slope geological conditions.

Generally, the number of parameters has little effect on neurons, and the number of parameters only represents the number of input neurons. In addition, the increase in number of parameters increases the simulation recognition time, and the actual engineering volume greatly increases. Therefore, to reduce the actual workload, this paper simplifies the input parameters of the model, and according to the modelling data and simulation results, the slope geological condition indicators are the freeze-thaw coefficient, hydrogeology, rock gravity, cohesion, internal friction angle, slope, slope height, porosity, and other factors.

(2) Set model output parameters

The output parameters are the grades of the geological conditions of the slopes in preparation for the iron ore mine, and the output parameters are divided into 4 grades according to the four expected output values of Grade I, Grade II, Grade III, and Grade IV. The specific content is shown in Table 1.

**Table 1.** Classification table of slope geological conditions.

| Geological Condition Level | Grade Description | Represents the Value |
|---|---|---|
| Grade I | Good, not easy to damage | (0, 0, 0, 1) |
| Grade II | Better, with potential destructive factors | (0, 0, 1, 0) |
| Grade III | Poor, damage may occur | (0, 1, 0, 0) |
| Grade IV | Poor, easy to cause damage | (1, 0, 0, 0) |

### 3.2. Determination of the Grid Structure

(1) Determination of the number of perceptrons

The input layer, hidden layer and output layer constitute the basic structure of the BP neural network. The number of hidden layers depends on the complexity of parameter selection. For a more complex problem to be solved, there are more hidden layers, and the difficulty of the corresponding model convergence increases.

(2) Determination of the number of network nodes

The method of dividing the network nodes of the input layer and output layer is unified and clear: generally, once the number of research projects is determined, the input layer and output layer are determined, but there is no scientific and consistent method of dividing the hidden layer of network nodes.

However, the neural network model constructed based on the slope parameters contains only a single hidden layer, so simply calculating the number of nodes in this layer can reveal the number of hidden layer nodes in the entire neural network model, which greatly simplifies the calculation process.

Because the number of rows of the input vector is equal to the number of nodes of the input layer, by knowing that the number of rows of the input vector is 8, it can be directly obtained that the input layer has 8 nodes. In addition, the number of nodes in the output layer is equal to the amount of output data points. Because the number of output data points is 4, it can be concluded that the number of nodes in the output layer is also 4. In addition to the above information and because the number of nodes in the hidden layer is 12, it is finally determined that the structure of the BP neural network is 8-12-4, as shown in Figure 5.

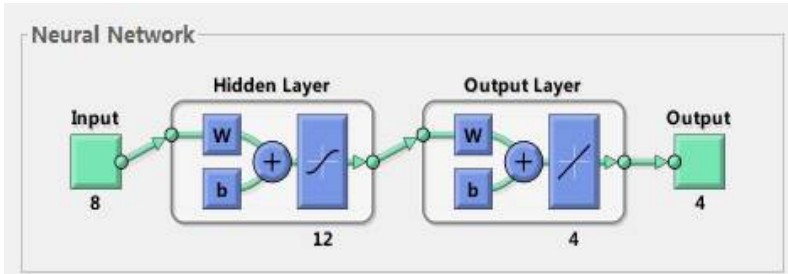

**Figure 5.** BP neural network structure diagram.

*3.3. Selection and Processing of Training Samples*

Two key factors of the network model training sample selection are mainly related to the complexity of the training sample. Firstly, the accuracy of the training sample. For training samples, the accuracy is positively correlated with the complexity of the samples. If the accuracy of the training samples increases, the complexity of the samples also increases, which will eventually increase the demand for the number of samples. Secondly, noise in the data. The noise in the sample data is also positively correlated with the complexity of the sample. If the noise in the data increases, the complexity of the training sample will also significantly increase, which affects the selection of the final training sample. Therefore, when training a neural network, it is necessary to control the relationship between the complexity of the sample and the accuracy of the training data and to provide as much key and useful information as possible to reduce the interference of redundant and useless information.

After consulting a large quantity of mine slope data, 54 neural network training samples were selected from them, with the literature [21,22] has mentioning the necessity of checking the dependency of each parameter before the ANN. However, the dependency of the parameters does not need to be discussed in this study. Because all the parameters are randomly selected, there is no dependence between the parameters in the nine main slope-influencing factors (according to the actual situation of Beizhan iron ore). All parameters are shown in Table 2, and the normalized data processing is shown in Table 3.

**Table 2.** Training sample parameter table.

| Serial Number | Freeze-Thaw Coefficient | Hydrology Geology | Unit Weight (KN/m$^3$) | Cohesion (KPa) | Internal Friction Angle φ (°) | Slope (°) | Slope Height (m) | Porosity (%) | Geology Grade |
|---|---|---|---|---|---|---|---|---|---|
| 1 | 0.42 | 2 | 12 | 0 | 30 | 45 | 8 | 1.62 | IV |
| 2 | 0.61 | 1 | 12 | 0 | 30 | 35 | 4 | 1.38 | II |
| 3 | 0.77 | 1 | 18 | 5 | 30 | 20 | 8 | 0.56 | I |
| 4 | 0.77 | 1 | 18 | 36 | 11 | 65 | 50 | 1.64 | I |
| 5 | 0.2 | 1 | 18.5 | 25 | 0 | 30 | 6 | 0.8 | IV |
| 7 | 0.42 | 2 | 20 | 20 | 36 | 45 | 50 | 1.38 | IV |
| 7 | 0.43 | 2 | 20 | 17 | 14 | 65 | 36 | 1.4 | III |
| 8 | 0.64 | 1 | 20 | 20 | 36 | 45 | 500 | 1.21 | IV |
| 9 | 0.76 | 1 | 21.4 | 10 | 30.34 | 30 | 20 | 0.65 | I |
| 10 | 0.62 | 1 | 21.4 | 8 | 28 | 45 | 31 | 0.73 | I |
| 11 | 0.68 | 2 | 21.4 | 10 | 30 | 30 | 20 | 0.75 | I |
| 12 | 0.54 | 1 | 22 | 10 | 36 | 45 | 50 | 1.1 | IV |
| 13 | 0.48 | 1 | 22 | 20 | 36 | 45 | 50 | 1.22 | IV |
| 14 | 0.33 | 2 | 22.4 | 10 | 35 | 45 | 10 | 1.62 | IV |
| 15 | 0.38 | 2 | 22.4 | 15 | 15 | 70 | 66 | 0.36 | I |
| 16 | 0.82 | 1 | 22.4 | 10 | 35 | 30 | 10 | 0.7 | I |
| 17 | 0.8 | 1 | 25 | 48 | 40 | 49 | 330 | 1.23 | I |
| 18 | 0.7 | 1 | 25 | 46 | 35 | 50 | 284 | 0.8 | II |
| 19 | 0.91 | 1 | 25 | 55 | 36 | 44.5 | 299 | 0.68 | I |
| 20 | 0.78 | 1 | 25 | 46 | 35 | 46 | 393 | 1.52 | I |
| 21 | 0.8 | 1 | 25 | 60 | 20 | 65 | 48 | 0.8 | IV |
| 22 | 0.7 | 1 | 25 | 20 | 16 | 45 | 123 | 1.3 | I |
| 23 | 0.91 | 1 | 25 | 50 | 35 | 50 | 84 | 0.66 | IV |
| 24 | 0.78 | 1 | 25 | 25 | 22 | 35 | 68 | 1.46 | IV |
| 25 | 0.4 | 2 | 26 | 150 | 45 | 30 | 200 | 1.46 | IV |
| 26 | 0.4 | 2 | 26 | 10 | 8 | 40 | 164 | 0.58 | I |
| 27 | 0.56 | 2 | 27 | 40 | 35 | 43 | 420 | 1.64 | IV |
| 28 | 0.88 | 1 | 27 | 50 | 40 | 42 | 407 | 0.8 | I |
| 29 | 0.93 | 1 | 27 | 35 | 35 | 42 | 359 | 0.68 | I |
| 30 | 0.35 | 2 | 27 | 32 | 33 | 42.4 | 289 | 1.4 | IV |
| 31 | 0.44 | 2 | 27 | 40 | 35 | 47.1 | 292 | 0.21 | IV |
| 32 | 0.84 | 1 | 27 | 37.5 | 35 | 37.8 | 320 | 0.65 | II |
| 33 | 0.36 | 2 | 27 | 17 | 20 | 50 | 98 | 0.56 | I |
| 34 | 0.55 | 1 | 27 | 16 | 13 | 60 | 164 | 0.68 | I |
| 35 | 0.88 | 2 | 27 | 18 | 45 | 70 | 212 | 0.82 | IV |
| 36 | 0.76 | 2 | 27 | 16 | 13 | 35 | 30 | 1.2 | IV |
| 37 | 0.37 | 1 | 27 | 17 | 20 | 80 | 15 | 0.96 | IV |
| 38 | 0.92 | 1 | 27.3 | 14 | 31 | 41 | 110 | 0.73 | II |
| 39 | 0.79 | 1 | 27.3 | 31.5 | 29.7 | 41 | 135 | 0.75 | I |
| 40 | 0.86 | 1 | 27.3 | 16.8 | 28 | 50 | 90.5 | 1.1 | III |
| 41 | 0.82 | 1 | 27.3 | 10 | 39 | 40 | 480 | 1.22 | I |
| 42 | 0.78 | 1 | 27.3 | 26 | 31 | 50 | 92 | 0.48 | I |
| 43 | 0.61 | 1 | 27.3 | 36 | 11 | 35 | 55 | 1.24 | I |
| 44 | 0.86 | 1 | 27.3 | 17 | 20 | 70.1 | 135 | 0.88 | IV |
| 45 | 0.54 | 1 | 27.3 | 60 | 23 | 45 | 95 | 0.92 | I |
| 46 | 0.46 | 1 | 27.3 | 14 | 17 | 45 | 22 | 0.66 | III |
| 47 | 0.56 | 2 | 31 | 68 | 37 | 49 | 200 | 0.68 | IV |
| 48 | 0.22 | 2 | 31.3 | 68 | 37 | 46 | 366 | 0.68 | IV |
| 49 | 0.47 | 2 | 31.3 | 68.6 | 37 | 47 | 305 | 1.52 | IV |
| 50 | 0.6 | 2 | 31.3 | 68 | 37 | 47 | 213 | 1.3 | IV |
| 51 | 0.22 | 2 | 31.3 | 20 | 15 | 30 | 35 | 1.4 | I |
| 52 | 0.47 | 2 | 31.3 | 14 | 17 | 60 | 22 | 0.86 | II |
| 53 | 0.33 | 2 | 31.3 | 5 | 34 | 55 | 10.5 | 1.23 | I |
| 54 | 0.74 | 2 | 31.3 | 60 | 25 | 52 | 143 | 0.76 | IV |

**Table 3.** Sample normalization.

| Serial Number | Freeze-Thaw Coefficient | Hydrology Geology | Unit Weight (KN/m³) | Cohesion (KPa) | Internal Friction Angle φ (°) | Slope (°) | Slope Height (m) | Porosity (%) |
|---|---|---|---|---|---|---|---|---|
| 1 | 0.971 | 0.911 | 0.467 | 1.000 | 0.333 | 1.000 | 0.644 | 0.928 |
| 2 | 0.965 | 0.943 | 0.314 | 1.000 | 0.714 | 1.000 | 0.771 | 0.921 |
| 3 | 0.986 | 0.970 | 0.185 | 0.698 | 1.000 | 0.321 | 0.495 | 1.000 |
| 4 | 1.000 | 0.993 | 0.463 | 0.097 | 0.681 | 1.000 | 0.533 | 0.973 |
| 5 | 0.987 | 0.933 | 0.233 | 0.667 | 1.000 | 1.000 | 0.600 | 0.947 |
| 7 | 1.000 | 0.941 | 0.213 | 0.213 | 0.433 | 0.799 | 1.000 | 0.967 |
| 7 | 1.000 | 0.957 | 0.396 | 0.489 | 0.584 | 1.000 | 0.100 | 0.975 |
| 8 | 1.000 | 0.999 | 0.922 | 0.922 | 0.858 | 0.822 | 1.000 | 0.998 |
| 9 | 0.993 | 0.976 | 0.398 | 0.370 | 1.000 | 0.977 | 0.303 | 1.000 |
| 10 | 1.000 | 0.983 | 0.064 | 0.667 | 0.234 | 1.000 | 0.369 | 0.995 |
| 11 | 1.000 | 0.910 | 0.413 | 0.364 | 1.000 | 1.000 | 0.318 | 0.995 |
| 12 | 1.000 | 0.981 | 0.132 | 0.617 | 0.434 | 0.798 | 1.000 | 0.977 |
| 13 | 1.000 | 0.979 | 0.131 | 0.212 | 0.435 | 0.798 | 1.000 | 0.970 |
| 14 | 1.000 | 0.925 | 0.012 | 0.567 | 0.552 | 1.000 | 0.567 | 0.942 |
| 15 | 0.999 | 0.953 | 0.367 | 0.580 | 0.580 | 1.000 | 0.885 | 1.000 |
| 16 | 0.993 | 0.983 | 0.265 | 0.458 | 1.000 | 0.708 | 0.458 | 1.000 |
| 17 | 1.000 | 0.999 | 0.853 | 0.713 | 0.762 | 0.707 | 1.000 | 0.997 |
| 18 | 1.000 | 0.998 | 0.828 | 0.680 | 0.758 | 0.652 | 1.000 | 0.999 |
| 19 | 0.998 | 0.998 | 0.837 | 0.636 | 0.763 | 0.706 | 1.000 | 1.000 |
| 20 | 1.000 | 0.999 | 0.876 | 0.769 | 0.826 | 0.769 | 1.000 | 0.996 |
| 21 | 1.000 | 0.994 | 0.246 | 0.844 | 0.402 | 1.000 | 0.470 | 1.000 |
| 22 | 1.000 | 0.995 | 0.603 | 0.684 | 0.750 | 0.276 | 1.000 | 0.990 |
| 23 | 0.994 | 0.992 | 0.416 | 0.184 | 0.175 | 0.183 | 1.000 | 1.000 |
| 24 | 1.000 | 0.994 | 0.280 | 0.280 | 0.370 | 0.017 | 1.000 | 0.9790 |
| 25 | 1.000 | 0.934 | 0.743 | 0.499 | 0.553 | 0.703 | 1.000 | 0.989 |
| 26 | 1.000 | 0.980 | 0.687 | 0.883 | 0.907 | 0.516 | 1.000 | 0.998 |
| 27 | 1.000 | 0.993 | 0.874 | 0.812 | 0.836 | 0.798 | 1.000 | 0.995 |
| 28 | 1.000 | 0.999 | 0.871 | 0.758 | 0.807 | 0.797 | 1.000 | 1.000 |
| 29 | 0.999 | 0.998 | 0.853 | 0.808 | 0.808 | 0.769 | 1.000 | 1.000 |
| 30 | 1.000 | 0.989 | 0.815 | 0.781 | 0.774 | 0.709 | 1.000 | 0.993 |
| 31 | 0.998 | 0.988 | 0.816 | 0.727 | 0.762 | 0.679 | 1.000 | 1.000 |
| 32 | 0.999 | 0.998 | 0.835 | 0.769 | 0.785 | 0.767 | 1.000 | 1.000 |
| 33 | 1.000 | 0.967 | 0.455 | 0.660 | 0.597 | 0.018 | 1.000 | 0.997 |
| 34 | 1.000 | 0.993 | 0.675 | 0.810 | 0.848 | 0.273 | 1.000 | 0.998 |
| 35 | 0.999 | 0.989 | 0.752 | 0.837 | 0.582 | 0.345 | 1.000 | 1.000 |
| 36 | 1.000 | 0.928 | 0.533 | 0.110 | 0.285 | 1.000 | 0.708 | 0.974 |
| 37 | 1.000 | 0.984 | 0.331 | 0.582 | 0.507 | 1.000 | 0.633 | 0.985 |
| 38 | 0.997 | 0.995 | 0.514 | 0.757 | 0.446 | 0.263 | 1.000 | 1.000 |
| 39 | 0.999 | 0.996 | 0.604 | 0.542 | 0.569 | 0.400 | 1.000 | 1.000 |
| 40 | 1.000 | 0.997 | 0.410 | 0.644 | 0.394 | 0.096 | 1.000 | 0.995 |
| 41 | 1.000 | 0.999 | 0.889 | 0.962 | 0.841 | 0.836 | 1.000 | 0.998 |
| 42 | 0.993 | 0.989 | 0.414 | 0.442 | 0.333 | 0.082 | 1.000 | 1.000 |
| 43 | 1.000 | 0.986 | 0.019 | 0.301 | 0.618 | 0.265 | 1.000 | 0.977 |
| 45 | 1.000 | 1.000 | 0.924 | 0.954 | 0.945 | 1.000 | 0.617 | 1.000 |
| 45 | 1.000 | 0.990 | 0.433 | 0.259 | 0.524 | 0.059 | 1.000 | 0.992 |
| 46 | 1.000 | 0.976 | 0.205 | 0.392 | 0.257 | 1.000 | 0.033 | 0.991 |
| 47 | 1.000 | 0.986 | 0.695 | 0.324 | 0.635 | 0.514 | 1.000 | 0.999 |
| 48 | 1.000 | 0.990 | 0.830 | 0.629 | 0.799 | 0.750 | 1.000 | 0.997 |
| 49 | 1.000 | 0.990 | 0.798 | 0.553 | 0.760 | 0.694 | 1.000 | 0.993 |
| 51 | 1.000 | 0.987 | 0.711 | 0.365 | 0.657 | 0.563 | 1.000 | 0.993 |
| 51 | 1.000 | 0.898 | 0.787 | 0.137 | 0.150 | 0.712 | 1.000 | 0.932 |
| 52 | 1.000 | 0.949 | 0.036 | 0.545 | 0.445 | 1.000 | 0.277 | 0.987 |
| 53 | 1.000 | 0.939 | 0.133 | 0.829 | 0.232 | 1.000 | 0.628 | 0.967 |
| 54 | 1.000 | 0.982 | 0.570 | 0.167 | 0.659 | 0.279 | 1.000 | 1.000 |

### 3.4. Sample Training and Result Analysis

The training steps are shown in Figure 6.

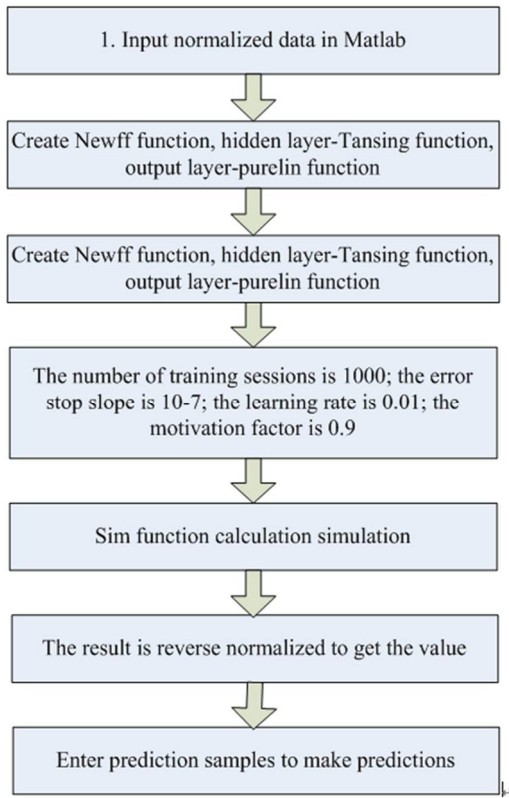

**Figure 6.** Training step diagram.

(1) Convergence graph

According to the data, the convergence curve is shown in Figure 7, which shows that the minimum momentum is added to the training so that the probability that the convergence curve has a local minimum is reduced after 1000 iterations of learning.

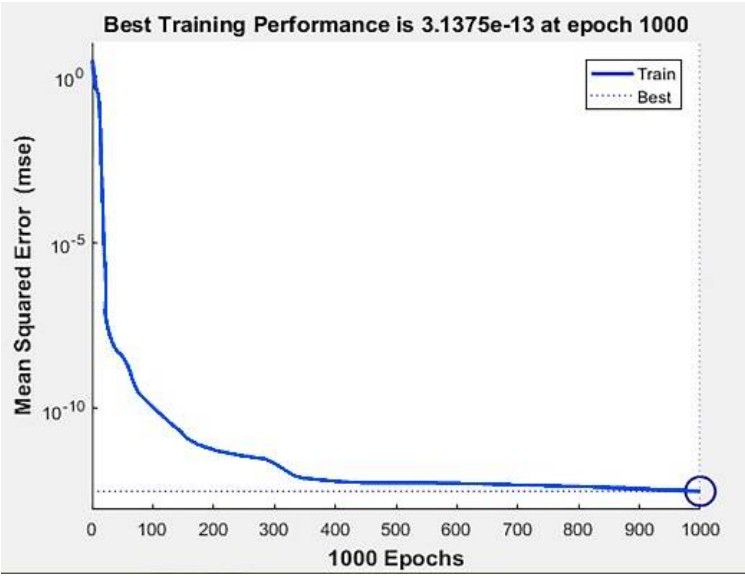

**Figure 7.** Convergent plot.

(2) Error distribution diagram

According to the data, the error distribution histogram is shown in Figure 8. The distribution histogram shows that the predicted sample is compared with the actual sample, and the error value is mostly distributed between –6% and 6%, which indicates that the result after training is reliable.

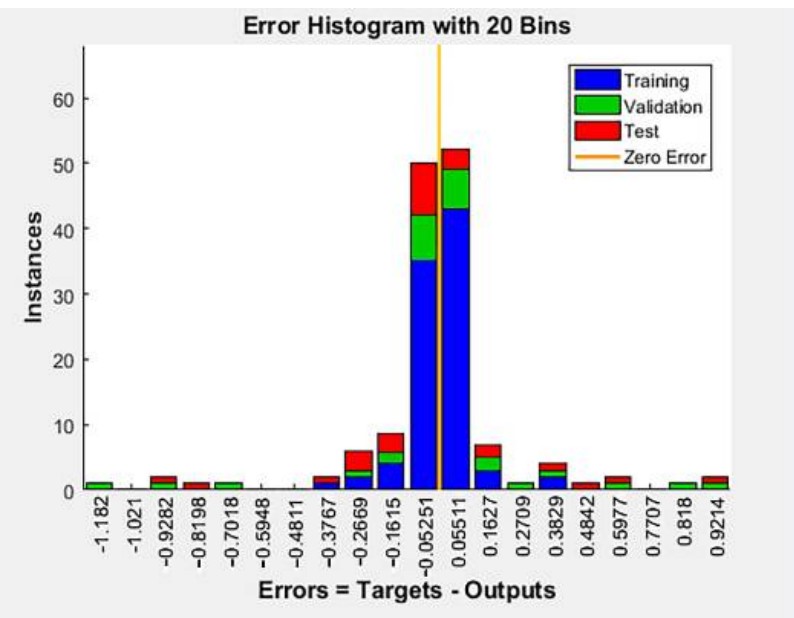

**Figure 8.** Error distribution histogram.

(3) Regression analysis graph

Regression diagrams are made according to the data: Figure 9 shows the regression diagram of 70% training samples; Figure 10 shows the regression diagram of 15% verification samples; Figure 11 shows the regression diagram of 15% test samples; Figure 12 shows the regression diagram of the overall sample. Among them, the abscissas 0 and 1 represent the target value, and the ordinate represents the sample value after debugging. If the slope of the curve approaches 1, it means that the target value is very close to the theoretical value, which implies that the regression analysis is very accurate.

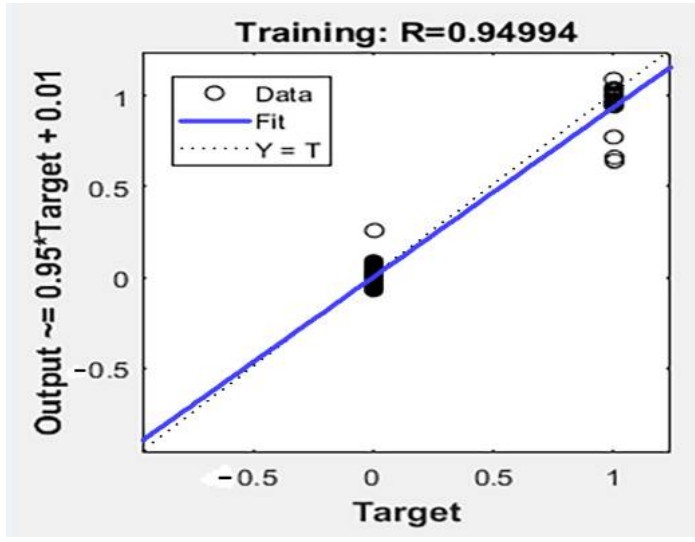

**Figure 9.** Regression diagram of training samples.

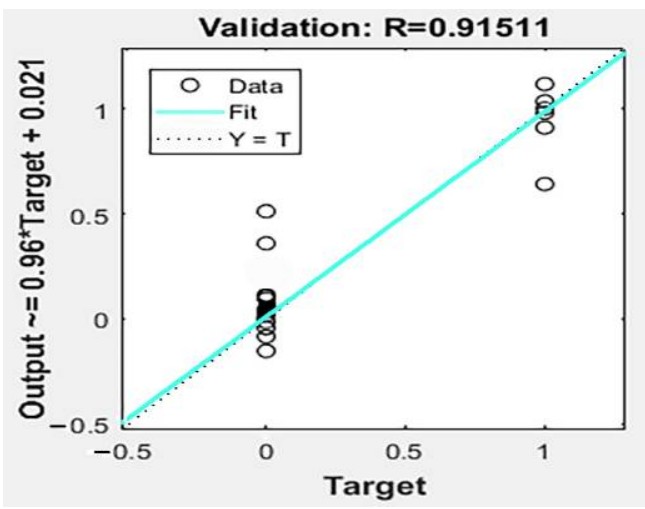

**Figure 10.** Regression graph of verify samples.

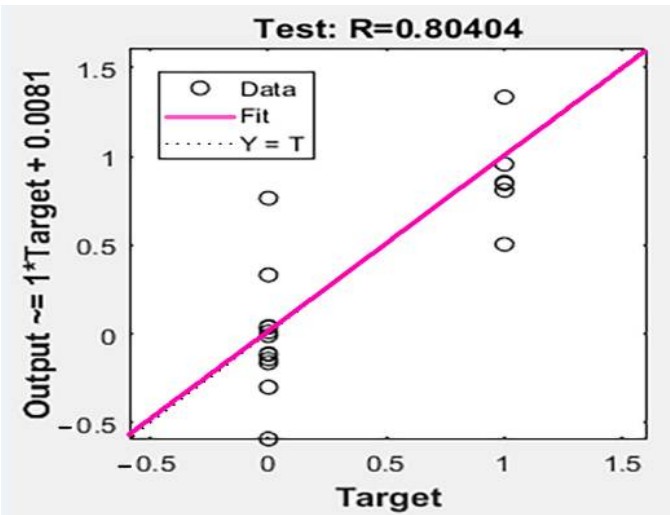

**Figure 11.** Regression diagram of test samples.

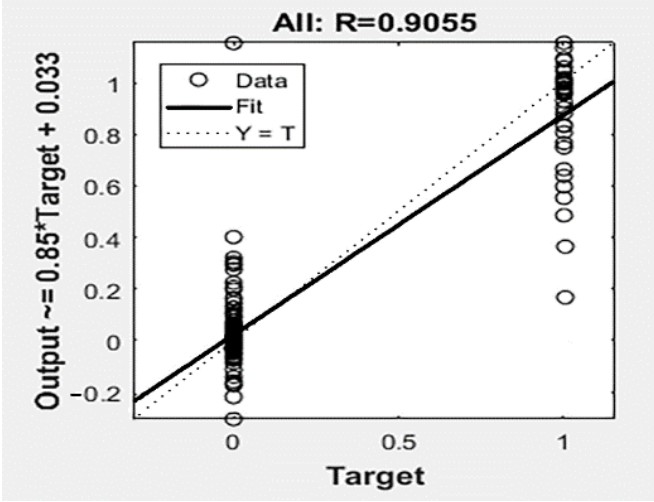

**Figure 12.** Regression diagram of overall samples.

The sample data in Table 4 represent the true value and predicted value of the sample and the error between them. In the table, the error is small (the maximum is 6.1%), which shows that the accuracy of network training is high.

**Table 4.** Comparison table of the actual situation and forecast results.

| Sample Number | Actual Value | | | | Predictive Value | | | | Error |
|---|---|---|---|---|---|---|---|---|---|
| 1 | 1 | 0 | 0 | 0 | 1.0323 | 0.0366 | 0.0356 | 0.0030 | 0.5% |
| 2 | 0 | 0 | 1 | 0 | 0.0524 | 0.1953 | 0.9481 | 0.0667 | 0.2% |
| 3 | 0 | 0 | 0 | 1 | 0.0081 | 0.2105 | 0.0158 | 0.9531 | 0.6% |
| 4 | 0 | 0 | 0 | 1 | 0.0758 | 0.2103 | 0.0406 | 0.9659 | 2.5% |
| 5 | 1 | 0 | 0 | 0 | 1.0478 | 0.2342 | 0.0339 | 0.2317 | 0.8% |
| 6 | 1 | 0 | 0 | 0 | 1.0489 | 0.0696 | 0.1896 | 0.0642 | 0.7% |
| 7 | 0 | 1 | 0 | 0 | 0.1145 | 1.0945 | 0.0374 | 0.0337 | 0.6% |
| 8 | 1 | 0 | 0 | 0 | 0.9673 | 0.3092 | 0.0422 | 0.0443 | 2.2% |
| 9 | 0 | 0 | 0 | 1 | 0.0918 | 0.0512 | 0.1962 | 1.0266 | 0.6% |
| 10 | 0 | 0 | 0 | 1 | 0.0752 | 0.0833 | 0.0209 | 1.0914 | 1.8% |
| 11 | 0 | 0 | 0 | 1 | 0.0293 | 0.2429 | 0.0194 | 0.9601 | 1.3% |
| 12 | 1 | 0 | 0 | 0 | 1.0302 | 0.2388 | 0.0558 | 0.0764 | 0.3% |
| 13 | 1 | 0 | 0 | 0 | 0.9711 | 0.2089 | 0.0480 | 0.3424 | 0.3% |
| 14 | 1 | 0 | 0 | 0 | 0.9865 | 0.0538 | 0.0670 | 0.1878 | 1.2% |
| 15 | 0 | 0 | 0 | 1 | 0.0209 | 0.2636 | 0.0014 | 0.9606 | 1.0% |
| 16 | 0 | 0 | 0 | 1 | 0.0847 | 0.0299 | 0.0434 | 0.9824 | 2.0% |
| 17 | 0 | 0 | 0 | 1 | 0.3618 | 0.2470 | 0.1938 | 0.9189 | 1.5% |
| 18 | 0 | 0 | 1 | 0 | 0.0325 | 0.0948 | 1.1287 | 0.0322 | 3.1% |
| 19 | 0 | 0 | 0 | 1 | 0.0057 | 0.0249 | 0.1771 | 0.9764 | 2.5% |
| 20 | 0 | 0 | 0 | 1 | 0.3223 | 0.2309 | 0.0500 | 1.0462 | 1.7% |
| 21 | 1 | 0 | 0 | 0 | 1.1517 | 0.1701 | 0.0186 | 0.4066 | 0.9% |
| 22 | 0 | 0 | 0 | 1 | 0.0044 | 0.0317 | 0.2256 | 1.1242 | 0.5% |
| 23 | 1 | 0 | 0 | 0 | 1.1481 | 0.2136 | 0.0531 | 0.0356 | 0.3% |
| 24 | 1 | 0 | 0 | 0 | 0.9817 | 0.0341 | 0.0399 | 0.0888 | 1.2% |
| 25 | 1 | 0 | 0 | 0 | 0.9511 | 0.2632 | 0.0954 | 0.3807 | 1.5% |
| 26 | 0 | 0 | 0 | 1 | 0.3125 | 0.0374 | 0.2708 | 1.0236 | 0.5% |
| 27 | 1 | 0 | 0 | 0 | 1.0945 | 0.1763 | 0.0556 | 0.1471 | 0.4% |
| 28 | 0 | 0 | 0 | 1 | 0.1113 | 0.2537 | 0.0143 | 0.9117 | 0.6% |
| 29 | 0 | 0 | 0 | 1 | 0.1085 | 0.2106 | 0.0484 | 0.2079 | 3.2% |
| 30 | 1 | 0 | 0 | 0 | 0.9495 | 0.1847 | 0.1203 | 0.3937 | 0.4% |
| 31 | 1 | 0 | 0 | 0 | 1.1025 | 0.2556 | 0.0029 | 0.2041 | 1.3% |
| 32 | 0 | 0 | 1 | 0 | 0.1315 | 0.0895 | 1.0778 | 0.1361 | 0.5% |
| 33 | 0 | 0 | 0 | 1 | 0.3289 | 0.3344 | 0.0241 | 0.2299 | 2.2% |
| 34 | 0 | 0 | 0 | 1 | 0.1908 | 0.0543 | 0.1349 | 0.9842 | 1.2% |
| 35 | 1 | 0 | 0 | 0 | 1.0539 | 0.0071 | 0.2508 | 0.3960 | 0.8% |
| 36 | 1 | 0 | 0 | 0 | 1.1564 | 0.0238 | 0.2560 | 0.3913 | 6.1% |
| 37 | 1 | 0 | 0 | 0 | 0.2317 | 0.0826 | 0.1567 | 0.0636 | 0.6% |
| 38 | 0 | 0 | 1 | 0 | 0.3396 | 0.0003 | 1.0133 | 0.0363 | 0.4% |
| 39 | 0 | 0 | 0 | 1 | 0.0323 | 0.0366 | 0.1356 | 1.0230 | 2.1% |
| 40 | 0 | 1 | 0 | 0 | 0.0524 | 0.9953 | 0.0481 | 0.0667 | 0.1% |
| 41 | 0 | 0 | 0 | 1 | 0.0081 | 0.0105 | 0.0158 | 0.9531 | 0.5% |
| 42 | 0 | 0 | 0 | 1 | 0.0758 | 0.2103 | 0.3406 | 0.9659 | 0.4% |
| 43 | 0 | 0 | 0 | 1 | 0.0511 | 0.0632 | 0.0954 | 0.9807 | 0.6% |
| 44 | 1 | 0 | 0 | 0 | 0.9825 | 0.0374 | 0.2708 | 0.0236 | 3.2% |
| 45 | 0 | 0 | 0 | 1 | 0.0945 | 0.1763 | 0.0556 | 0.9471 | 0.4% |
| 46 | 0 | 1 | 0 | 0 | 0.1113 | 1.0537 | 0.0143 | 0.2117 | 1.3% |
| 47 | 1 | 0 | 0 | 0 | 1.1085 | 0.2106 | 0.0484 | 0.2079 | 0.5% |
| 48 | 1 | 0 | 0 | 0 | 0.9495 | 0.1847 | 0.1203 | 0.3937 | 2.2% |
| 49 | 1 | 0 | 0 | 0 | 1.1025 | 0.2556 | 0.0029 | 0.2041 | 1.2% |
| 50 | 1 | 0 | 0 | 0 | 1.1315 | 0.0895 | 0.0778 | 0.1361 | 0.8% |
| 51 | 0 | 0 | 0 | 1 | 0.0289 | 0.3344 | 0.0241 | 0.9299 | 0.6% |
| 52 | 0 | 0 | 1 | 0 | 0.0308 | 0.0543 | 0.9349 | 0.0842 | 0.8% |
| 53 | 0 | 0 | 0 | 1 | 0.0539 | 0.0071 | 0.2508 | 0.9960 | 1.6% |
| 54 | 1 | 0 | 0 | 0 | 0.9564 | 0.0238 | 0.0560 | 0.3913 | 2.3% |

## 4. Grade Division of Slope Geological Conditions in Preparation for Iron Mines

*4.1. Determination of Parameter Samples of Geological Condition Indicators*

There are two main mining areas in the current mining area of the preparation for iron ore: open-pit mining and side-hanging mining. The slope area of this study is mainly the side slope between the pit and the side-hanging mine, which forms after the open-pit mining, i.e., the east side slope area of the mine. Referring to the geological report of the area and the index data of the test items, a set of parameter samples containing geological conditions index can be obtained, as shown in Table 5.

**Table 5.** Indicators and parameters of slope geological conditions.

| Sample | Freeze-Thaw Coefficient | Hydrogeology | Unit Weight (kN/m$^3$) | Cohesion C (MPa) | Internal Friction Angle $\varphi$ (°) | Slope Gradient (°) | Slope Height (m) | Porosity (%) |
|---|---|---|---|---|---|---|---|---|
| 1 | 0.88 | 2 | 25 | 8.2 | 28.8 | 65 | 122 | 1.96 |
| 2 | 0.83 | 2 | 23.7 | 7.3 | 31 | 27 | 185 | 1.25 |
| 3 | 0.76 | 2 | 28.4 | 18.2 | 28.3 | 28 | 137 | 0.7 |
| 4 | 0.92 | 2 | 24.1 | 11.1 | 29.6 | 37 | 240 | 1.23 |
| 5 | 0.9 | 2 | 24.8 | 3.2 | 37.9 | 36 | 185 | 0.8 |
| 6 | 0.94 | 2 | 29.2 | 17.7 | 33.3 | 38 | 180 | 0.68 |
| 7 | 0.36 | 1 | 25.3 | 6.8 | 30.6 | 55 | 80 | 1.94 |
| 8 | 0.28 | 1 | 24.2 | 4.5 | 35.1 | 60 | 85 | 0.8 |
| 9 | 0.7 | 1 | 23.8 | 4.3 | 32.4 | 52 | 40 | 1.65 |
| 10 | 0.72 | 1 | 27.2 | 11.1 | 30.4 | 31 | 73 | 0.73 |
| 11 | 0.75 | 1 | 26.4 | 9 | 31 | 35 | 30 | 0.75 |
| 12 | 0.8 | 1 | 27 | 12.3 | 33 | 41 | 55 | 1.1 |
| 13 | 0.79 | 1 | 29.6 | 8.5 | 32.2 | 43 | 35 | 1 |

*4.2. Calculation Results and Analysis*

According to the training results of the BP neural network model based on the training samples in the previous section, the accuracy of the network is high, so it can be used to prepare for the calculation of the iron ore geological index parameter samples. After normalizing the data in the geological condition parameter table (Table 5), it is input into the neural network model for calculation, and the result is shown in Table 6. Table 7 is obtained after summarizing the samples of the same geological condition level among 13 groups of samples.

**Table 6.** Classification table of slope geological conditions.

| Sample Number | MATLAB Algorithm Prediction Results | | | | Slope Grade |
|---|---|---|---|---|---|
| 1 | 0.0759 | 0.0378 | 0.0457 | 1.0286 | I |
| 2 | 0.1556 | 0.9543 | 0.1673 | 0.1744 | III |
| 3 | 1.0885 | 0.0844 | 0.2401 | 0.9332 | I |
| 4 | 0.1281 | 0.3648 | 0.3191 | 0.1930 | IV |
| 5 | 0.3929 | 0.4178 | 0.1265 | 0.7675 | II |
| 6 | 0.4028 | 0.3458 | 0.1476 | 0.9204 | I |
| 7 | 0.5088 | 0.0963 | 0.8371 | 0.2608 | II |
| 8 | 0.3147 | 0.4021 | 0.3597 | 0.3112 | I |
| 9 | 0.5993 | 0.1380 | 0.8891 | 0.2342 | IV |
| 10 | 0.1224 | 0.0445 | 0.9446 | 0.1536 | I |
| 11 | 0.2659 | 0.0408 | 0.6098 | 0.1285 | III |
| 12 | 0.5896 | 0.2179 | 0.7807 | 0.1128 | III |
| 13 | 0.5292 | 0.1991 | 0.7404 | 0.1057 | II |

**Table 7.** Summary of the grades of the slope geological conditions.

| | Grade and Status of Slope Geological Conditions |
|---|---|
| 1, 3, 6, 8, 10 | Grade I: good geological conditions, not easy to damage |
| 5, 7, 13 | Grade II: Good geological conditions, with potential damage factors |
| 2, 11, 12 | Grade III: The geological conditions are poor, which may cause damage |
| 4, 9 | Grade IV: Poor geological conditions, easy to cause damage |

Based on Table 7, the BP neural network analysis shows that among the 13 samples of the eastern slope in this cold area, 5 have geological conditions of Grade I, and 3 have geological conditions of Grade II. There are 3 with condition Grade III and 2 with Grade IV conditions. The 13 sample numbers are distributed in different locations on the eastern slope. After their positions have been marked on the eastern slope, the distribution area map of the eastern slope samples, as shown in Figure 13, is obtained.

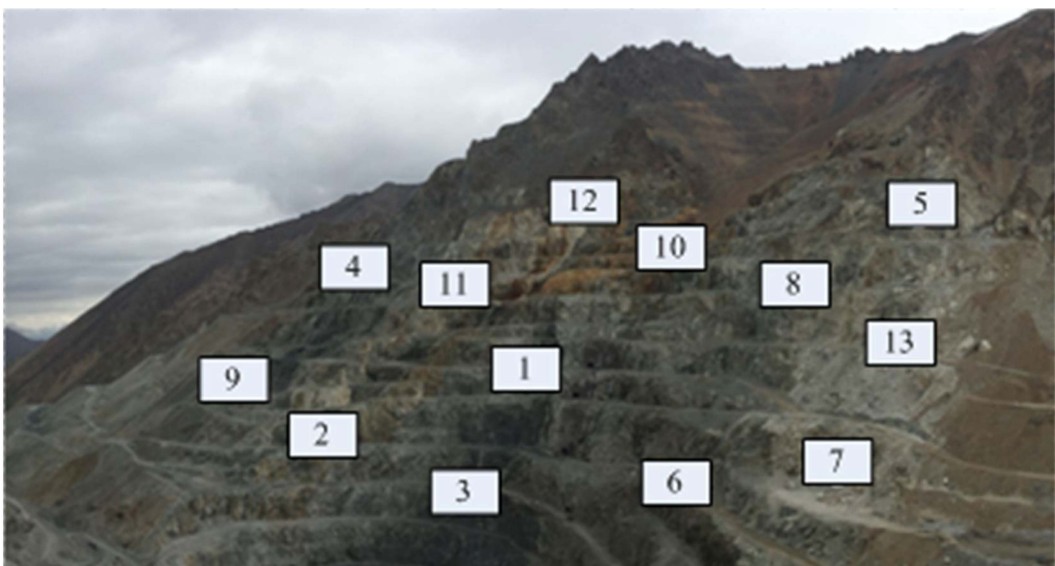

**Figure 13.** Sample distribution map of the east slope.

The numbers 1 to 13 in the diagram are the sampling points, and 13 samples are taken from different areas of the east slope of Beizhan Iron Mine. Figure 13 shows that although the distribution positions of the 13 samples on the eastern slope are random; there is a certain distribution law, i.e., the distribution among the samples at identical or similar geological condition levels is relatively dense, and samples of different geological conditions are far apart and sparsely distributed. As a result, the regions where samples with identical or similar geological condition levels are located can be statistically divided, so that the eastern slope can be divided into geological conditions. The specific divisions are shown in Figure 14.

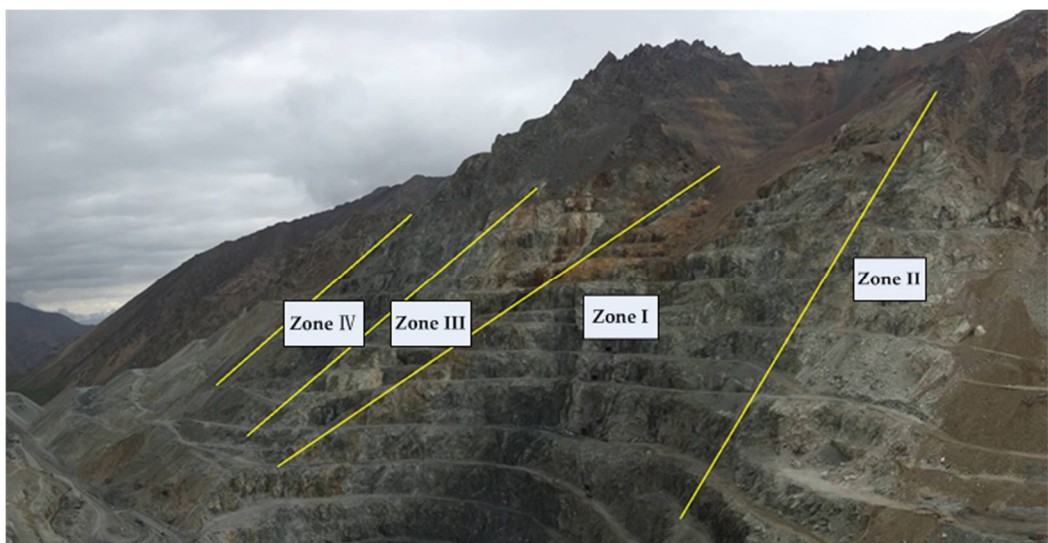

**Figure 14.** Zoning map of geological conditions of the east slope.

### 5. Concluding Remarks

A BP neural network is used to classify the geological conditions of the eastern slope of the preparatory iron mine and the overall division. The eastern slope of the iron mine preparation is divided into four areas: Zone I, Zone II, Zone III, and Zone IV. The corresponding geological condition grades for zones I to IV are grade I, grade II, grade III, and grade IV, respectively. Among them, the rock formation in Zone I is mainly skarn rock formation, which is also the main occurrence area of ore bodies. It has high unit weight, high hardness, undeveloped rock joints, high integrity, and good physical and mechanical properties, so its geological conditions are good. Damage does not easily occur, and the corresponding geological condition is grade I. The rock formation in Zone II is mainly monzonite porphyry. Compared with skarn its weight and hardness are slightly lower, however, the rock layer is thick and the joints are less developed. Therefore, it has better physical and mechanical properties. The conditions are good, there are only potential destructive factors, and the corresponding geological conditions are grade II. The rock formations in Zone III are mainly marble formations. Compared with skarn and monzonite porphyries, marble is relatively poor in lithology, has low gravity and hardness, and has more joints in the formations. The physical and mechanical properties are poor, but its thickness is large, and the layered distribution slightly compensates for the lack of lithology. Therefore, its geological conditions are general, and there is a possibility of damage. The corresponding geological conditions are grade III. The rock formation in Zone IV is mainly limestone rock. It has the worst lithology among the four rock formations, with low unit weight, low hardness, well-developed joints, and large porosity. After long-term weathering, erosion, and freezing and thawing cycles, its physical properties are destroyed. Therefore, the geological conditions in this area are poor and easily destroyed. The corresponding geological conditions are grade IV.

**Author Contributions:** The research articles with four authors, the Conceptualization R.Z. and S.W.; methodology, Q.C.; software, S.W; validation, R.Z., S.W and Q.C.; formal analysis, C.X.; investigation, C.X.; resources, Q.C.; data curation, S.W.; writing—original draft preparation, S.W.; writing—review and editing, R.Z.; visualization, C.X.; supervision, Q.C.; project administration, Q.C. All authors have read and agreed to the published version of the manuscript.

**Funding:** The research was funded by the national key research and development project "Slope instability mechanism and early warning technology of open pit in high altitude and cold area" (No. 2018YFC0808402) and the Hunan Province Science Foundation, grant number 2021JJ30679.

**Institutional Review Board Statement:** This paper does not involve human or animal studies.

**Informed Consent Statement:** This paper does not involve human research.

**Data Availability Statement:** All the data included in this study are available upon request by contact with the corresponding author.

**Conflicts of Interest:** The authors declare no conflict of interest.

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
