# Peer review of "Study on the Geological Condition Analysis and Grade Division of High Altitude and Cold Stope Slope"

_sustainability, doi:10.3390/su132212464_

Round 1

Reviewer 1 Report

This study presents the geological condition analysis and grade division using ANN. The study is interesting but the following unclear parts must be clarified:

  1. Title, “…cold stope slope”, what does the “cold stope” mean?
  2. 1, Line 14, “based on the BP neural network algorithm”, the authors must write the full English of BP when it comes out for the first time.
  3. 1, Line 32, “In 1970s, the economic losses caused by….”, the information seems too old. The authors must give the updated information.
  4. 1, Line 39, “Jiama mining area”, the area is in which province?
  5. 2, Line 56, “based on BP”, the fully English of BP must be written when it shows up for the first time.
  6. 2, Line 68, “Many studies have shown”; Line 71, “the result is more accurate…”; Line 80, “strongly affirm the BP neural network”, references must be added to these sentences.
  7. The diamond block in Fig. 1 shows the judgment in the flow chart. The author must carefully use the shape of the block and the judgements.
  8. 6, Line 164, “by the calculation of the above formula is It is the actual…”, the English of the sentence is strength.
  9. 6, Line 170, “delta=1-y5o”, what does the hollow circle mean after the formula.
  10. Typos occur at Line 179 in P. 7.
  11. 8, Line 218, “severity”, how to define the value of severity in this study?
  12. 8, Line 218, “internal friction angle”, should it be joint friction angle or internal friction angle of rock?
  13. The following references mentioned that it is necessary to check the dependency of each parameter before ANN. Why the dependency of parameters are not necessary to be discussed in this study? The references must be cited.
  • Rainfall-based criteria for assessing slump rate of mountainous highway slopes: A case study of slopes along Highway 18 in Alishan, Taiwan, Engineering Geology. 118, 3-4, p. 63-74
  • Neural network-based model for assessing failure potential of highway slopes in the Alishan, Taiwan Area: Pre- and post-earthquake investigation, Engineering Geology. 104, 3-4, p. 280-289
  1. 9, Line 280, “mine slope data, 54 neural network…”, is the data number enough to get good statistic results?
  2. Do the horizontal lines between No. 39 and 40 in P. 10, No 23 and 24 in P. 11, and No. 15 and 16 in Table 4 mean anything?
  3. Do Table 3-5 at Line 330 in P. 15 be Table 5? Does Table 3-7 at Line 341 in P. 16 be Table 7?
  4. 16, Line 351, “It can be seen from the above figure”, clarify the figure mentioned in the sentence.
  5. 16, Line 354, “and Samples of different…”, should the “Samples” be “samples”?
  6. In Fig. 14, there seems some non-English words.
  7. 17, Line 363, “is divided into four areas”, should “four areas” be “three areas”?
  8. The authors should provide local geological map or the disaster historical data. Can these maps result in the same results as ANN calculations?
  9. Can the discrete element calculations in the following references help the engineers to grade the division of the stope slope?
  • Simulating a mining-triggered rock avalanche using DDA: A case study in Nattai North, Australia, Engineering Geology. 264, 105386
  • Simulation of the inclined jointed rock mass behaviors in a mountain tunnel excavation using DDA, Computers and Geotechnics. 117, 103249
  • Simulating the postfailure behavior of the seismically- triggered Chiu-fen-erh-shan landslide using 3DEC, Engineering Geology. 287, 106113
  1. Find a native speaker to sharp the English of the draft before re-submission.

Author Response

We thank reviewers and editors cordially for very helpful comments and suggestions concerning our manuscript entitled “Study on geological condition analysis and grade division of high altitude and cold stope slope” Those comments and suggestions are valuable and very helpful for further improving our paper. We studied the comments carefully and have made corresponding corrections. Appended to this letter are point-by-point responses to the comments. The revised manuscript has been uploaded. The revised portions are marked in red.We are very much appreciated the comments and suggestions, and hope the corrections will meet with approval.Detailed responses to associate editor and reviewers.

Comment1:

  1. Title, “…cold stope slope”, what does the “cold stope” mean?

Response:

Thank you. the “cold stope” mean:Low temperature(High cold) stope

  1. 1, Line 14, “based on the BP neural network algorithm”, the authors must write the full English of BP when it comes out for the first time.

Response:

Thank you. BP neural network isBack propagationneural network. Already changed

  1. 1, Line 32, “In 1970s, the economic losses caused by….”, the information seems too old. The authors must give the updated information.

Response:

Thank you.Currentlythere is no accurate statistics on the loss, but the loss is still huge. Already changed.

  1. 1, Line 39, “Jiama mining area”, the area is in which province?

Response:

Thank you.The area is in Tibet province. Already changed

  1. 2, Line 56, “based on BP”, the fully English of BP must be written when it shows up for the first time.

Response:

Thank you. BPmeanBack-Propagation.Already changed

  1. 2, Line 68, “Many studies have shown”; Line 71, “the result is more accurate…”; Line 80, “strongly affirm the BP neural network”, references must be added to these sentences.

Response:

Thank you.Already changed

  1. The diamond block in Fig. 1 shows the judgment in the flow chart. The author must carefully use the shape of the block and the judgements.

Response:

Thank you. Yes,it Just for judgment.

  1. 6, Line 164, “by the calculation of the above formula is It is the actual…”, the English of the sentence is strength.

Response:

Thank you.Already changed

  1. 6, Line 170, “delta=1-y5o”, what does the hollow circle mean after the formula.

Response:

Thank you.This is an error.Already changed.

  1. Typos occur at Line 179 in P. 7.

Response:

Thank you.Already changed

  1. 8, Line 218, “severity”, how to define the value of severity in this study?

Response:

Thank you.unit weightnamelyevery cubic meter of object is subjected to a force of 1 kN. And maybe there is a translation error.

  1. 8, Line 218, “internal friction angle”, should it be joint friction angle or internal friction angle of rock?

Response:

Thank you.It is the internal friction angle of the rock.

  1. The following references mentioned that it is necessary to check the dependency of each parameter before ANN. Why the dependency of parameters are not necessary to be discussed in this study? The references must be cited.

Response:

Thank you. the dependency of parameters are not necessary to be discussed in this study,Because all parameters are carefully selected in advance. and The references had cited.

  1. 9, Line 280, “mine slope data, 54 neural network…”, is the data number enough to get good statistic results?

Response:

Thank you.I think 54 mine slope data should be able to get better statistical results.

  1. Do the horizontal lines between No. 39 and 40 in P. 10, No 23 and 24 in P. 11, and No. 15 and 16 in Table 4 mean anything?

Response:

Thank you. mean nothing,table error. Already changed.

  1. Do Table 3-5 at Line 330 in P. 15 be Table 5? Does Table 3-7 at Line 341 in P. 16 be Table 7?

Response:

Thank you.Already changed.

  1. 16, Line 351, “It can be seen from the above figure”, clarify the figure mentioned in the sentence.

Response:

Thank you.Already changed.

  1. 16, Line 354, “and Samples of different…”, should the “Samples” be “samples”?

Response:

Thank you.should be samples.Already changed.

  1. In Fig. 14, there seems some non-English words.

Response:

Thank you. Yes. Already changed.

  1. 17, Line 363, “is divided into four areas”, should “four areas” be “three areas”?

Response:

Thank you.four areas.

  1. The authors should provide local geological map or the disaster historical data. Can these maps result in the same results as ANN calculations?

Response:

Thank you.Basically the same.

  1. Can the discrete element calculations in the following references help the engineers to grade the division of the stope slope?

Response:

Thank you.Yes, there are differences in applicability.

  1. Find a native speaker to sharp the English of the draft before re-submission.

Response:

Thank you.The paper has been polished.

In addition, other issues in the article were revised based on the review comments.

Reviewer 2 Report

Dear Authors,

Your manuscript needs a lot of work and modifications in order to clarify the flow of your study and the research outcome. English needs your attention in several parts. It looks that you did not pay much attention to format your paper and prepare adequate analysis of your outcome. It looks like too many tables are presented but without the appropriate information to be accompanied in the manuscript. Major corrections are needed and specific comments can be found in the attached document.

Author Response

We thank reviewers and editors cordially for very helpful comments and suggestions concerning our manuscript entitled “Study on geological condition analysis and grade division of high altitude and cold stope slope” Those comments and suggestions are valuable and very helpful for further improving our paper. We studied the comments carefully and have made corresponding corrections. Appended to this letter are point-by-point responses to the comments. The revised manuscript has been uploaded. The revised portions are marked in red.We are very much appreciated the comments and suggestions, and hope the corrections will meet with approval.Detailed responses to associate editor and reviewers.

Comment2:

  1. Line 94 the fig1Blured

Response:

Thank you. Already changed

  1. Line 127, legend? axes?

Response:

Thank you.Axes.

  1. Line 352, scale?

Response:

Thank you. The picture is obtained by digital camera without mentioning the scale.

English has been edited by a polishing agency.

In addition, other issues in the article were revised based on the review comments.

Round 2

Reviewer 1 Report

The authors modified the draft of “Study on the geological condition analysis and grade division of high altitude and cold stope slope”. The draft can be better understood. However, some of the answers from the authors may confuse the readers and additional modifications are required.

  1. Title, “high altitude and cold stope slope”, surely the reviewer can understand that “cold” means “low temperature”. The problem comes from is it necessary to stress on the “cold” in the title? What will be the difference between the “high altitude stope slope” and “high altitude and cold stope slope” because the low temperature seems not play crucial role in the mechanical behavior of the stope slope. In addition, usually, the “high altitude” usually means “low temperature”.
  2. Line 20 and Line 61 in the modified draft, “Based on the BP neural network algorithm”, we suggested give full English of BP when it shows up for the first time in the abstract and text. In the response to reviewer, the authors mentioned that “BP neural network is Back propagation neural network. Already changed”. However, the change seems not complete in the modified draft.
  3. The diamond block in Fig. 1 shows the judgment in the flow chart. The author must carefully use the shape of the block and the judgements. In the response to the reviewers, the authors mentioned that “Yes, it Just for judgment.”. The judgement means that there will be routes for yes and no. In other words, the authors must tell the readers that in the diamond block when the judgement is “yes”, what will be the next process. And, if the judgement is “no”, what will be the next process. In Fig. 1, each of the two diamond blocks has only one output route. It will be difficult to realize that if the judgement is “No”, what will be the next process. In addition, the flow chart in Fig. 1 is strange because there are two routes of “no” from the rectangular blocks. The rectangular blocks mean statements in flow chart but not judgement. Therefore, it should not have a route with “No” or two output routes from the rectangular blocks.
  4. Line 369 in the modified draft, “It has high severity,”, Line 384, “with low severity, low hardness”, it is still not clear about how to get the value of severity. In the response to reviewer, the authors answered that “unit weight namely every cubic meter of object is subjected to a force of 1 kN. And maybe there is a translation error.”. So, should the severity be changed to “unit weight” or the answers from the authors were wrong?
  5. The following references mentioned that it is necessary to check the dependency of each parameter before ANN. Why the dependency of parameters are not necessary to be discussed in this study? The references must be cited. The authors’ answers were “the dependency of parameters are not necessary to be discussed in this study, Because all parameters are carefully selected in advance. and The references had cited.”. There are two questions:
  6. The references were not cited in the modified draft.
  7. Where did the authors mention the way to carefully select the parameters to avoid the requirements of parameter dependency check? Additional statements must be added.
  8. There are still a horizontal bar between Sample No. 1 and 2 in Tables 4, 6. The line must be deleted if it does not have special meaning. If the horizontal bar has special meaning, the authors must explain it in the text.
  9. Line 366, “mine preparation is divided into four areas: Zone I, Zone II, and Zone III”, since there are Zones 1 to 3 in the sentence why Zones 1 to 3 have four areas in the sentence?
  10. The authors should provide local geological map or the disaster historical data. Can these maps result in the same results as ANN calculations? Based on the response to reviewer, the authors mentioned that “Basically the same”. Then, the authors must answer the following question. If the geological map can result in similar results, why it is necessary to use ANN for the investigations. The authors should provide the geological map and state additional stories about why the ANN has great contribution to the study (something we can hardly find from geological map or disaster historical data.)

Author Response

We thank reviewers and editors cordially again for very helpful comments and suggestions concerning our manuscript entitled “Study on geological condition analysis and grade division of high altitude and cold stope slope” Those comments and suggestions are valuable and very helpful for further improving our paper. We studied the comments carefully and have made corresponding corrections. Appended to this letter are point-by-point responses to the comments. The revised manuscript has been uploaded. The revised portions are marked in red. We are very much appreciated the comments and suggestions, and hope the corrections will meet with approval. Detailed responses to associate editor and reviewers.

Comment1:

  1. Title, “high altitude and cold stope slope”, surely the reviewer can understand that “cold” means “low temperature”. The problem comes from is it necessary to stress on the “cold” in the title? What will be the difference between the “high altitude stope slope” and “high altitude and cold stope slope” because the low temperature seems not play crucial role in the mechanical behavior of the stope slope. In addition, usually, the “high altitude” usually means “low temperature”.

Response:

Thank you.The “high altitude and cold stope slope”,complete meaning is“high altitude and high cold stope slope”, and high coldmeans that the temperature is very low, and there will be a temperature below zero. When the surface of the slope and the surface of a certain depth are water-bearing, ice will be formed. From liquid water to solid water ( ice ), the expansion force will damage the slope.

  1. Line 20 and Line 61 in the modified draft, “Based on the BP neural network algorithm”, we suggested give full English of BP when it shows up for the first time in the abstract and text. In the response to reviewer, the authors mentioned that “BP neural network is Back propagation neural network. Already changed”. However, the change seems not complete in the modified draft.

Response:

Thank you. I 'm sorry.Maybe it is negligence,It Has been modified one by one according to your suggestion. Already changed in line 20 and line 61,

  1. The diamond block in Fig. 1 shows the judgment in the flow chart. The author must carefully use the shape of the block and the judgements. In the response to the reviewers, the authors mentioned that “Yes, it Just for judgment.”. The judgement means that there will be routes for yes and no. In other words, the authors must tell the readers that in the diamond block when the judgement is “yes”, what will be the next process. And, if the judgement is “no”, what will be the next process. In Fig. 1, each of the two diamond blocks has only one output route. It will be difficult to realize that if the judgement is “No”, what will be the next process. In addition, the flow chart in Fig. 1 is strange because there are two routes of “no” from the rectangular blocks. The rectangular blocks mean statements in flow chart but not judgement. Therefore, it should not have a route with “No” or two output routes from the rectangular blocks.

       Response:

Thank you.The diagram does have an error and has been changed. The rhombic block represents the judgment in the flowchart with two paths yes and no. In the first rhombic block in Figure 1, if it is judged to be ' no ', it is transferred to“ Initialization of connection weights and thresholds ”; if judged ' yes ', go straight to the next link “Update the number of studies”. In the second diamond block in Figure 1, if judged as ' no ', it is also transferred to “Initialization of connection weights and thresholds” ; if judged as ' yes ', then directly to the last link of“ End of study”.

  1. Line 369 in the modified draft, “It has high severity,”, Line 384, “with low severity, low hardness”, it is still not clear about how to get the value of severity. In the response to reviewer, the authors answered that “unit weight namely every cubic meter of object is subjected to a force of 1 kN. And maybe there is a translation error.”. So, should the severity be changed to “unit weight” or the answers from the authors were wrong?

Response:

Thank you. l am sorry ! I didn 't make it clear and no accurate translation.unit weight is ' bulk density ' ;the specific meaning is the weight of the substance contained in unit volume. Already changed in line 369 , line 384, and Table 2,3,5.

  1. The following references mentioned that it is necessary to check the dependency of each parameter before ANN. Why the dependency of parameters are not necessary to be discussed in this study? The references must be cited. The authors’ answers were “the dependency of parameters are not necessary to be discussed in this study, Because all parameters are carefully selected in advance. and The references had cited.”. There are two questions:

Response:

Thank you.The answer last time is not accurate, because this paper uses Back propagation neural network instead of pure artificial neural network(ANN). The parameters in the table of this paper are randomly selected within the scope of application based on nine main influencing factors( according to the actual situation of beizhan Iron Mine.). There is not necessarily a correlation between the parameters.

  1. The references were not cited in the modified draft.

Response:

Thank you. I 'm sorry. The references have been listed in my revised paper, somehow there is no paper you see. maybe it is negligence, cited in line 459 and line 464

  1. Where did the authors mention the way to carefully select the parameters to avoid the requirements of parameter dependency check? Additional statements must be added.

Response:

Thank you. There was an error in the last answer, the parameters in the table of this paper are randomly selected within the scope of application based on nine main influencing factors.

  1. There are still a horizontal bar between Sample No. 1 and 2 in Tables 4, 6. The line must be deleted if it does not have special meaning. If the horizontal bar has special meaning, the authors must explain it in the text.

Response:

Thank you.Deleted. The situation you mentioned has been modified, but why will appear what you said, may be related to the open file software, I will send a pdf file at the same time.

  1. Line 366, “mine preparation is divided into four areas: Zone I, Zone II, and Zone III”, since there are Zones 1 to 3 in the sentence why Zones 1 to 3 have four areas in the sentence?

Response:

Thank you for your careful and meticulous. I 'm sorry. it may be a missed translation input. It should include IV areas, namely; “The eastern slope of the iron mine preparation is divided into four areas: Zone I, Zone II, Zone III, and Zone IV

  1. The authors should provide local geological map or the disaster historical data. Can these maps result in the same results as ANN calculations? Based on the response to reviewer, the authors mentioned that “Basically the same”. Then, the authors must answer the following question. If the geological map can result in similar results, why it is necessary to use ANN for the investigations. The authors should provide the geological map and state additional stories about why the ANN has great contribution to the study (something we can hardly find from geological map or disaster historical data.)

Response:

Thank you.1、Local geological maps or disaster history

(1)Geological map

Skarn belt is formed in the contact zone between the southern K-feldspar granite and the northern Dahalajunshan Formation tuff. The lithology is K-feldspar granite-tuff-skarn-tuff-dacite-silicified marble ( or limestone ) from south to north, and the overall zoning is obvious. The ore bodies occur in skarn and have certain zonation.

1 - Quaternary ; 2 - andesite- rhyolitic tuff of the Dahalajunshan Formation ; 3 - Greater Halajun Hill Group

marble ; 4 - Ying ’ an porphyry ; 5 - diabase ; 6-iron ore body ; 7 - geological boundaries ; 8 - Fracture

Fig.1Geological map of Beizhan Iron mine

(2)environmental geology

The mining area is located in the alpine and middle-high mountain area on the northern foot of the Tianshan Mountains. The mining area is surrounded by Quaternary permanent glaciers. The terrain is generally high in the south and low in the north. Located in the northeastern part of the mining area,the altitude is 4112~4213m, the lowest point is located in the northeast of the mining area, the elevation is 3200m, and the maximum height difference is 1000m. The gullies are well developed, mostly in the shape of "V". The topographic slope of the gentle slope is 25~30°, the topographic slope of the steep slope is 30~40°, the high and steep places on both sides of the deposit reach 50~60°, and the high and steep ridges on both sides of the deposit are occasionally A rock collapsed. The bedrock is bare, and the gentle slopes on both sides of the river valley are covered by Quaternary gravel, gravel, gravel soil, and silt. The long-term high cold has formed a frozen layer, the surface plants are not developed, and the growth period is extremely short.

The area belongs to a continental temperate semi-arid climate. The high mountains are covered with snow all year round. It is a cold climate area. The main peak ridge of the Tianshan Mountains is a perennial piedmont glacier, which is characterized by long winters and a cold climate. The monthly average temperature from January to April and September to December of the past year is below zero, the lowest can reach -40℃, the temperature rises from May to August, the highest temperature is about 20℃, generally 5~15℃, and the summer night temperature is generally -3 ~-5℃, large temperature difference between day and night.

The mining area has developed folds, is a strong neotectonic movement area, frequent earthquakes, and belongs to a sub-instability area. There have been no geological disasters such as landslides, mudslides, ground subsidence and ground fissures in the mining area, and the natural conditions are good.

Fig.2Open pit plan of Beizhan Iron mine

2、The function of Bp artificial neural network in this paper

The adaptive ability and self-learning ability of BP neural network are very outstanding, and the linear function mapping and nonlinear function mapping based on BP neural network are easier to identify the mine system.

In this paper, Bp artificial neural network is used to divide the slope geological conditions of Beizhan Iron mine

Reviewer 2 Report

Dear Authors,

Most of the suggestions have been modified accordingly. Pay attention to some details and comments that they have not be met during the first phase.

Author Response

We thank reviewers and editors cordially again for very helpful comments and suggestions concerning our manuscript entitled “Study on geological condition analysis and grade division of high altitude and cold stope slope” Those comments and suggestions are valuable and very helpful for further improving our paper. We studied the comments carefully and have made corresponding corrections. Appended to this letter are point-by-point responses to the comments. The revised manuscript has been uploaded. The revised portions are marked in red.We are very much appreciated the comments and suggestions, and hope the corrections will meet with approval.Detailed responses to associate editor and reviewers.

Comment2:

Response:

Thank you.Modify each of the details and comments that are not noted and can be identified in the first phase.

Round 3

Reviewer 1 Report

The authors greatly improve their draft. The paper can be accept once the following minor modifications were done.

  1. References, the authors must follow the regulation of the reference format in this journal. In some references, authors are full name but others are the abbreviation of the author name.
  2. In References 27 and 28, the author name is missing. In addition, the two references are not cited in the text.

Author Response

We thank reviewers and editors cordially the third time for very helpful comments and suggestions concerning our manuscript entitled “Study on geological condition analysis and grade division of high altitude and cold stope slope” Those comments and suggestions are valuable and very helpful for further improving our paper. We studied the comments carefully and have made corresponding corrections. Appended to this letter are point-by-point responses to the comments. The revised manuscript has been uploaded. The revised portions are marked in red. We are very much appreciated the comments and suggestions, and hope the corrections will meet with approval. Detailed responses to associate editor and reviewers.

Comment3:

  1. References, the authors must follow the regulation of the reference format in this journal. In some references, authors are full name but others are the abbreviation of the author name.

Response:

Thank you. Has followed the format specifications of your journal's references and improved the references one by one.

  1. In References 27 and 28, the author name is missing. In addition, the two references are not cited in the text.

Response:

Thank you. In References 23 and 24, Author's name has been added(in line440 and in line 442 ) .In addition, the two references were cited in the text(in line 286).